# A Comparison of the Contents of Disaster Nursing Practices and Perceived Difficulties among Nurses Working at Welfare Evacuation Shelters during Natural Disasters and Multiple Disasters: A Qualitative Study

**DOI:** 10.3390/ijerph192416610

**Published:** 2022-12-10

**Authors:** Yoshiko Shiomitsu, Takumi Yamaguchi, Keiko Imamura, Tamami Koyama, Hitomi Tsuchihashi, Yuta Kawaoka, Yuko Matsunari

**Affiliations:** 1School of Health Sciences, Kagoshima University, Kagoshima 890-0075, Japan; 2Department of Nursing, Faculty of Nursing and Nutrition, Kagoshima Immaculate Heart University, Kagoshima 895-0011, Japan; 3Center for Preventive Medical Sciences, Chiba University, Chiba 263-8522, Japan; 4Radiation Emergency Medicine Research Center, Nuclear Safety Research Association, Tokyo 105-0004, Japan; 5School of Nursing, Tsuruga Nursing University, Fukui 914-0814, Japan

**Keywords:** disaster nursing, welfare evacuation shelters, natural disasters, nuclear disasters, Fukushima Daiichi nuclear power station accident, public health nursing, disaster relief operations

## Abstract

In this study, we compared the disaster relief practices of nurses who worked in welfare shelters in Iwate and Miyagi Prefectures, areas in which only natural disasters occurred, and nurses who worked in Fukushima Prefecture, an area in which both nuclear and natural disasters occurred during the Great East Japan Earthquake in 2011, in order to identify commonalities and differences between them. We conducted semi-structured interviews with two nurses from each prefecture. The results revealed that “nursing practice with minimal available materials and personnel” and “nursing practice based on knowledge and experience as a nurse” were common themes in the content of nursing practices, whereas “securing human resources during disasters and considering ideal welfare evacuation centers” and “recording the difficulties in dealing with nuclear disasters” were uncommon themes. The findings confirmed that even in Fukushima Prefecture, in which the nuclear disaster occurred, participants did not talk about their concerns regarding radiation exposure while working at welfare evacuation shelters where people with special requirements were evacuated, and that they were expected to respond in the same way as they would in natural disasters. However, participants reported several difficulties relating to nuclear disasters that should be considered in future disaster support.

## 1. Introduction

On 11 March 2011, the Great East Japan Earthquake (GEJE; the largest earthquake recorded in Japan to date) occurred, and was accompanied by a tsunami induced by the earthquake [1,2]. The earthquake and tsunami affected the Tohoku and Kanto regions; over 10,000 houses were fully destroyed and over 12,000 houses were partially destroyed [1]. Furthermore, 15,853 deaths were caused by the disaster, with 6013 people injured and 3286 missing around the affected area as of 8 February 2011 [3]. Additionally, an accident at the Fukushima Daiichi Nuclear Power Station (FDNPS) occurred following the earthquake and tsunami. As a result of the accident, large amounts of radionuclides were released into the atmosphere and dispersed onto the ground, contaminating the environment. Many residents of Fukushima Prefecture were forced to evacuate their hometown. The evacuees included elderly people, people with disabilities, children, and women [4]. Elderly people and people with disabilities living at nursing facilities often require medical and nursing care at all times, and many were unable to receive care because of the emergency situation. As a result, many evacuees, particularly older people and those with disabilities, died during evacuation [5,6]. These types of fatalities have been referred to as “disaster-related deaths.” There were 929 and 469 disaster-related deaths in Iwate and Miyagi Prefectures, respectively, which were the areas most severely affected by the earthquake and tsunami. However, 2313 disaster-related deaths occurred in Fukushima Prefecture, constituting the highest number of any prefecture [5]. This situation occurred because patients who lived in nursing homes and hospitals in Fukushima Prefecture were forced to evacuate because of an evacuation order by the Japanese government to avoid radiation exposure, which treated the patients the same as able-bodied persons. Mortality rates among patients were reported to increase as a result of the evacuation [7]. The situation regarding evacuation in Fukushima Prefecture was different from that of any other prefecture. Therefore, it is likely that differences in nursing care and associated difficulties occurred between Fukushima Prefecture, Iwate Prefecture, and Miyagi Prefecture. 

In Japan, multiple disasters occur every year. When disasters occur, the primary victims are elderly people and people with disabilities, who are referred to as “people requiring special care”. The Japanese government has been establishing “welfare evacuation shelters” for these groups since 1995, when the Great Hanshin-Awaji Earthquake occurred in Japan [8]. The cabinet office of the government of Japan reported that welfare evacuation shelters have been established each time a serious disaster has occurred; however, the response system and the supply of professional relief were not sufficient in the GEJE [9]. A previous study regarding infection in shelters in Iwate and Miyagi Prefectures at GEJE reported that a large number of evacuees were exposed to cold, unhygienic conditions, and malnutrition because of power failures, insufficient food provision, and a lack of running water at evacuation centers after the GEJE, putting the evacuees, particularly people requiring special care, at an increased risk of contracting infectious diseases [10]. Because the GEJE was not only a natural disaster but also a nuclear disaster, it is likely that many nurses involved in disaster relief at welfare evacuation shelters in Fukushima Prefecture experienced difficulties. However, to date, no studies have compared the contents or difficulties involved in disaster relief nursing at welfare evacuation shelters between situations involving a natural disaster alone and situations involving both a natural disaster and a nuclear disaster. 

Clarifying the differences in nursing practices between nuclear and natural disasters may be useful for informing nursing activities in the event of future nuclear disasters. In the present study, therefore, we aimed to clarify the nursing practices undertaken at welfare evacuation shelters in Iwate and Miyagi Prefectures, where only natural disasters occurred, and in Fukushima Prefecture, where a nuclear disaster also occurred in addition to a natural disaster, and examined the differences in nursing practices. 

## 2. Materials and Methods

### 2.1. Study Design

This was a qualitative study using semi-structured interviews.

### 2.2. Study Participants

First, we sent information documents explaining the research and response sheets for study participation to nurses working at departments in the prefectures and municipalities of interest, as well as community comprehensive care centers, designated welfare evacuation shelters and other facilities, activity groups, and individuals (nurses) at welfare evacuation shelters. Nurses who did not undertake nursing activities in disaster nursing during the GEJE were excluded from the study. A total of 91 nurses responded to our response sheet. Of these, 78 nurses declined to participate in the interview, and 13 nurses were willing to participate in the study. Of these, seven nurses were excluded because they were busy with daily nursing practices, and their schedule was not adjusted. Therefore, six nurses were finally enrolled in our study. Our final participants included two nurses working in Iwate Prefecture, two nurses working in Miyagi Prefecture, and two nurses working in Fukushima Prefecture at the time of the disaster (Figure 1).

### 2.3. The Contents of the Interview Guide

i.The process of nursing practice during the disaster and the disaster relief situation.
·What department(s) did you belong to? What kind of work were you responsible for?·Please tell me about the process from the occurrence of the earthquake to the nursing practice at the welfare evacuation shelter and the period of activity. ·What do you feel about the activities at the welfare evacuation shelter?·Please give an overview of the welfare evacuation shelter at which you practiced. ·Please tell me about the roles and contents of the disaster relief you were involved in. ii.Nurses’ experiences of disaster relief
·What diseases were exhibited by the elderly people you assisted?·What symptoms were exhibited by the elderly people you assisted?·What other assistance was required? 
iii.Effective nursing practices and problems
·Please tell me about nursing practices that seemed to work well in the midst of a shortage of goods, people, and information.·Please tell me about nursing practices that seemed to not work well in the midst of a lack of knowledge and equipment. ·Please tell me specifically about, e.g., the physical resources, human resources, and information that were in short supply. 
iv.Future disaster relief
·Please describe any advice you have for nurses who will implement nursing practices at welfare evacuation shelters in the future. 

### 2.4. Data Collection

The interviews were conducted from April to June 2018 by a trained interviewer (Y.S., the faculty of nursing university) in the participants’ workplace or home. The interviewer followed a semi-structured interview guide. The duration of the interview was approximately 1 h per person. The contents of the interview were recorded with an Integrated Circuit recorder, and the data were transcribed for analysis. 

### 2.5. Data Analysis

The interviews were analyzed using qualitative content analysis [11,12]. Coding was performed by marking condensed meaning units of the transcribed text for each interview. Thereafter, we gathered codes that resembled meaning and created categories. Regarding content analysis, to compare the contents of disaster nursing practices and difficulties, we integrated the interview data into two sections: a group affected by a natural disaster alone, and a group affected by multiple disasters. All of the analyzing processes (i.e., coding and categorizing processes) were implemented by all co-authors with discussions to increase reliability and validity. 

We translated Japanese into English after the content analysis because the interview data were collected in Japanese. In the translation procedure, we carefully confirmed that the meaning did not change from the original version. 

## 3. Results

Demographic data of the study participants are shown in Table 1. All participants were female and participants’ ages ranged from people in their 40s to those in their 60s. In addition, the years of nursing experience ranged from 24 to 42 years. 

Table 2 shows the results of a comparison of category names between “Iwate Prefecture and Miyagi Prefecture” and “Fukushima Prefecture.” 

### 3.1. Process of Disaster Nursing Practices and Disaster Relief Situations

Regarding this content theme, similar results were obtained in Miyagi, Iwate, and Fukushima Prefectures, in which the categories “Nursing was performed with available supplies” and “Nursing during a shortage of supplies and personnel” were extracted, respectively. An example of an actual narrative is as follows: 


*(No. 1) It also became necessary to apply common sense, or rather, the ingenuity of applying what was around us to make up for the goods we did not have.*



*(No. 2) There were desks, chairs, and stationery available, and it was easy because there were resources available for each plan.*


### 3.2. Experience among Nurses in Disaster Relief and Effective Nursing Practices and Problems

This item was also common to all groups, with “nursing to the best of my knowledge and experience as a nurse” being extracted from this item. An example of an actual narrative is as follows:


*(No. 1) In receiving evacuees from elderly care facilities, we also changed nasal catheters and indwelling bladder catheters, suctioned tracheas, and administered injections and IV fluids where facilities were available. However, some facilities provided specialized nursing care such as monitoring, insulin administration, wound care, pressure ulcer treatment, AED use, and postmortem care. In addition, experienced nurses were available to quickly identify patients who needed to be addressed.*



*(No. 5) Public Health Nurses have little direct experience in nursing care or long-term care, so rather than being good at that, they are always looking at people and thinking about what is needed now and what can be done to prevent it in the future, and what can be returned to many people.*


### 3.3. For Future Disaster Relief

Regarding this type of content, the category generated for Iwate and Miyagi Prefectures (areas that experienced only natural disasters) was “Securing human resources during disasters and considering ideal welfare evacuation centers,” whereas the category generated for Fukushima Prefecture, which also experienced a nuclear disaster, was “Recording the difficulties in dealing with disasters.” The category selected for Fukushima Prefecture, which also experienced a nuclear disaster, was “Recording the difficulties in dealing with nuclear disasters.” An example of an actual narrative is as follows:


*(No. 4) There were people with disabilities among able-bodied people. When I was making my rounds on patrol, I thought, “This person does not look well,” so I called out to him and peeked in. This person possibly had a mental or intellectual disability, and was unable to successfully communicate that he was not feeling well.*



*(No. 5) In a sense, there are living teaching materials nearby in Fukushima, so I think we should examine what happened and make use of them to determine how areas with nuclear power plants should respond in the event of an accident.*


## 4. Discussion

In the present study, we interviewed nurses who worked in Iwate, Miyagi, and Fukushima Prefectures and performed disaster relief at welfare evacuation shelters that had adopted disaster nursing practices in welfare evacuation shelters. Additionally, we compared disaster relief performed by nurses in Iwate and Miyagi Prefectures, which experienced natural disasters, with those in Fukushima Prefecture, which experienced both natural and nuclear disasters, as well as their thoughts regarding disaster nursing in the future. The number of nurses who cooperated with the interview survey was extremely limited. This allowed us to finally reach the target population and get a general idea of their nursing practices. At the time of the GEJE, welfare evacuation centers themselves were set up hastily as secondary evacuation centers, and the system and environment were not yet in place, so many people did not have the confidence to tell others about their activities. We assumed this is the reason why many of them regretted what they should have done. Therefore, it was also assumed that nurses with relatively long experience and confidence in their nursing practices were interviewed.

### 4.1. Consistencies Regarding Disaster Relief among Nurses in Iwate and Miyagi Prefectures and Nurses in Fukushima Prefecture

The categories generated as commonalities in disaster relief operations were “nursing practice with minimal available materials and personnel” and “nursing practice based on knowledge and experience as a nurse.” Regarding the destruction of houses in the GEJE, it is reported that approximately 120,000 houses were completely destroyed, approximately 280,000 houses were half destroyed, and approximately 730,000 houses were partially destroyed [13]. The tsunami also affected many homes, and the transportation of supplies from outside the prefecture came to a halt. Therefore, it was necessary to provide nursing care with minimal supplies. Generally, at disaster sites, distribution of supplies stops, supplies are lost, or equipment breaks down because of disasters. Therefore, it is necessary for medical personnel to substitute medical and nursing supplies with what is available on site. Because the participants in the present study had many years of nursing experience, they appeared to be able to practice nursing in a large-scale disaster based on their previous experience. However, this does not mean that all nurses who respond to large-scale disasters have a long duration of experience; newly-appointed nurses are also required to respond in disaster situations. Disaster nursing practice was not common among the current participants, and was conducted on the basis of empirical rules. Therefore, more practical disaster nursing education is needed [14,15]. We assume that the skills required to consider and decide what can be substituted for necessary supplies are based on previous nursing experience. Therefore, regardless of whether or not a nuclear disaster occurred, “nursing practice with minimal available materials and personnel” and “nursing practice based on knowledge and experience as a nurse” were considered to be commonly practiced.

One possible reason for the absence of differences in nursing practices between Iwate and Miyagi Prefectures, in which only natural disasters occurred, and Fukushima Prefecture, in which both natural and nuclear disasters occurred, is that welfare evacuation shelters were places in which elderly people, disabled people, and others requiring special care took refuge. It has been reported elsewhere that nurses felt helpless because they were unable to respond to the many consultations regarding radiation [16]. However, the nurses working at the welfare evacuation shelters in Fukushima Prefecture in the current study did not report that concerns about radiation were raised by disaster victims, but rather talked about their efforts to provide the best possible nursing care to the patients in front of them. This may explain why the residents who were evacuated to the welfare evacuation shelters did not express concerns about radiation, even in Fukushima Prefecture.

### 4.2. Differences in Disaster Support among Nurses in Iwate and Miyagi Prefectures and Nurses in Fukushima

In Iwate and Miyagi Prefectures, in which only natural disasters occurred, the number of deaths and missing persons were higher than those in Fukushima Prefecture, and nursing care had to be provided under difficult conditions such as shortages of medical supplies and food. In particular, because many people requiring special care were staying in welfare evacuation shelters, a higher level of nursing practice was required. Therefore, if disaster nursing practices differ depending on the length of experience as nurses, equal nursing care may not be provided, and lives that could be saved may be lost. In this sense, it is highly likely that the nurses recognized the need to secure human resources.

In Fukushima, because of the nuclear disaster, the category “Recording the difficulties in dealing with nuclear disasters” was generated, and this point was mentioned by various information sources after the nuclear disaster. Certainly, nursing practices within welfare evacuation shelters will remain the same whether a natural disaster or a concurrent nuclear disaster occurs, and nursing practices are based on previous experience and knowledge with minimal materials. However, it is likely that anxiety about radiation caused a change in perceptions outside of the on-site response. The situation regarding radiation was not as severe as it seemed at the time. It has been reported that the perception of radiation as a high risk was related to the mental health of nurses working in hospitals in Fukushima Prefecture [17], and that radiation anxiety was also related to intention to leave the workforce [18]. Therefore, it may be necessary for many nurses to learn about nursing practices related to nuclear disasters and the health effects of the Fukushima accident, in order to reinforce their mental state and mental health; this is not typically included in the content of nursing practices.

### 4.3. Strengths and Limitations

This is the first study to compare the narratives of nurses who worked in welfare evacuation shelters in prefectures that experienced only natural disasters with those of nurses who worked in welfare evacuation shelters in prefectures that experienced not only natural disasters but also nuclear disasters, and to identify consistencies and differences among them. However, several limitations were observed in the present study. First, because the study participants were all nurses with many years of experience, we were unable to examine the perceptions of younger nurses about the difficulties they experienced. However, because nurses who worked in the welfare shelters had relatively long durations of experience, they were able to cover a wide range of nursing practices. Second, the number of nurses in each prefecture in which interviews were conducted was small. At the time this study was conducted, the effects of the GEJE were still lingering, and it was not possible to obtain research consent from active nurses. However, we were able to interview nurses who were in charge of welfare evacuation shelters at the time of the earthquake, and to obtain their narratives from a broader perspective. However, future surveys should include qualitative research using alternative types other than interview surveys to support the small number of subjects. Third, because this was a qualitative study, the results may not reflect an overall trend. We plan to conduct further studies to address these limitations in the future. Furthermore, this survey was conducted with nurses who were working in welfare shelters at the time of the GEJE, and no survey was conducted with users of welfare shelters. In order to examine the future of welfare shelters, it was decided that a future task would be to conduct an interview survey of users of welfare shelters as well.

## 5. Conclusions

Comparison of the activities of nurses who worked in welfare shelters in Iwate and Miyagi Prefectures, in which only natural disasters occurred during the GEJE, and nurses who worked in welfare shelters in Fukushima Prefecture, in which nuclear disasters occurred in addition to natural disasters, suggested that “nursing practice with minimal available materials and personnel” and “nursing practice based on knowledge and experience as a nurse” were common themes. Nursing practices in welfare shelters had low possibility to include any content related to the nuclear disaster. In contrast, regarding future disasters, nurses from Iwate and Miyagi Prefectures reported the need to “secure human resources during disasters and consider ideal welfare evacuation centers,” whereas nurses from Fukushima Prefecture reported the need to “recording the difficulties in dealing with nuclear disasters.” It is possible that nurses in Fukushima Prefecture perceived radiation as a high risk not only for the patients they treated directly, but also for all people affected by the unprecedented disaster that occurred around them, and this may be the reason for the differences in the content of their responses. Although various practices of disaster nursing education have already been conducted [19], it is highly likely that the disaster nursing education that has been conducted so far will be effective regardless of the location of disaster nursing practices or the type of disaster, and not limited to welfare shelters, which are the subject of this study.

## Figures and Tables

**Figure 1 ijerph-19-16610-f001:**
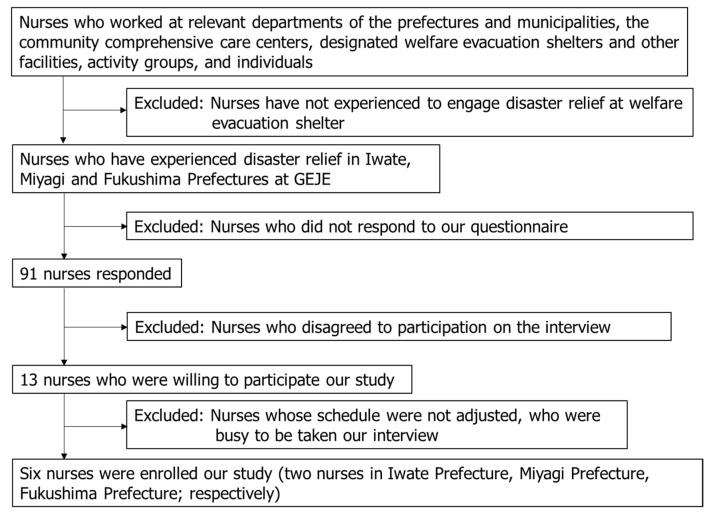
Enrollment of study participants.

**Table 1 ijerph-19-16610-t001:** Characteristics of the study participants.

No.	Sex	Present Affiliation	Age	Employment	Nursing Career	Area	Place of Interview
1	Female	N/A (Volunteer)	60s	Registered nurse	42 y	Iwate	House
2	Female	Nursing home	50s	Registered nurse	34 y	Iwate	Workplace
3	Female	N/A (Volunteer)	40s	Registered nurse	24 y	Miyagi	Workplace
4	Female	Municipality	50s	Public health nurse	33 y	Miyagi	Workplace
5	Female	Municipality	50s	Public health nurse	35 y	Fukushima	Workplace
6	Female	Nursing home	40s	Registered nurse	29 y	Fukushima	Workplace

**Table 2 ijerph-19-16610-t002:** Comparison of disaster nursing contents and situation from each interview.

	Iwate and Miyagi Prefectures (Affected Only by Natural Disaster; No. 1–4)		Fukushima Prefecture (Affected by Multiple Disasters, Including Nuclear Disaster; No. 5, 6)	
Contents	Categories	Codes	Categories	Codes
Process of disaster nursing practice, disaster relief situation	Nursing was performed with available supplies.	At the shelter, the place was transformed from a place for health checkups to a place for talking about the disaster situation and what was on their mind. Nursing support was provided at the request of busy public health nurses. The services included health care for residents, support for activities of daily living, coordination with other facilities, and providing a place for healing. Nurses stayed on standby in case communication was disrupted and to respond to any problems that might arise. As a calm environment for the evacuees, beds were placed in the dining hall, and for family evacuees, a conference room was used as a living space where they could rest. The shelter operation was a mixture of a general shelter and a welfare evacuation shelter, with both functioning 24 h a day. Nursing care at the welfare shelter consisted primarily of checking vitals because lifelines had not returned, and there was no place to provide nursing skills. Accompanied by a traveling doctor.	Nursing during a shortage of supplies and personnel.	Because of a lack of water, cold weather, and an inability to prepare large quantities of warm food, maintaining cleanliness and assisting with toileting were difficult. There were many areas that could not be handled, and we looked for a place that could accommodate people in a facility or in a short period of time. In the maintenance of the living environment at the evacuation shelter, power transmission was suspended, and heating could not be maintained, so stoves were rented to maintain the heating. Transportation and relocation of severely ill patients who had difficulty receiving medical treatment were undertaken, as well as coordination of continued support because the number of staff was decreased by the evacuation order.
Experience among nurses in disaster relief	Nursing care that fully utilizes your experience and knowledge as a nurse.	The nurses’ work required them to judge the condition of the evacuees, apply their experience, and provide appropriate support. For each person requiring care, it was necessary to judge whether it was better to administer medical attention or to request a Disaster Medical Assistance Team to examine them. With almost no information provided by the evacuees, nurses had to ascertain their condition. To manage evacuees who were in psychological distress, nurses spoke to them while taking care to use appropriate language. The environment was maintained with an emphasis on infection prevention, medication management, and safe prescriptions even without a medical record. Because there were situations that required management of bed placement and self-determination, we created and compiled medical records for those in need of assistance. Blood pressure measurements, knowledge of medications, and emergency procedures were applied for evacuees who were being treated for diabetes.	Nursing care that fully utilizes your experience and knowledge as a nurse.	The roles of the public health nurses and registered nurses differed, with the public health nurses providing coordination and assessment, and the registered nurses implementing content related to medical intervention. In the absence of a medical doctor, nurses performed medical treatment to the best of their ability. Assistance at the shelter was to distribute medications and help prevent people from falling over while toileting during the nighttime hours.
Effective nursing practices and problems	Nursing practices for victims on the basis of experience and knowledge.	The support provided depended on the facility environment of the welfare evacuation shelter, the nurses’ years of experience, and whether nurses were public health nurses. It is necessary to listen to what the survivors themselves have to say, but nurses did not have time to listen to them carefully. With the passage of time after the disaster, it became necessary to move from life-saving support to maintaining cleanliness and giving people a sense of peace of mind.	Nursing practices on the basis of previous experience.	Although few public health nurses had direct nursing care experience, they could predict what was needed in the current situation. It is necessary to respond to difficulties in food, clothing, and shelter for living, to deal with transportation difficulties, and to secure living space. The experience of working in the past was put to good use after the disaster, and there was no panicking. By gathering all people requiring special care in one place, a system was created to enable early detection of abnormalities based on their experiences.
For future disaster relief	Securing human resources during disasters and considering ideal welfare evacuation centers.	It is important to make use of potential nurses and to identify areas for improvement in daily life management. In the future, we will examine the operations and state of welfare shelters. Flexible use should be made of goods and support. To secure human resources, we should enhance helper training. Multidisciplinary cooperation is important. Centers should be operated as a combined welfare evacuation shelter and first aid station. It should be ensured that there are no shelters where evacuees with intellectual or mental disabilities are not differentiated from general evacuees.	Recording the difficulties in dealing with nuclear disasters.	Regarding nuclear disasters, Fukushima should be utilized as a case study. Nurses who engaged in disaster relief will make use of proposals for Japan’s disaster system, the current situation and future of Fukushima, problems associated with the nuclear accident, analysis of information and its dissemination, and the struggle against radiation anxiety.

## Data Availability

All data are available from the corresponding author on reasonable request.

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
