# Peer review of "A Comparison of the Contents of Disaster Nursing Practices and Perceived Difficulties among Nurses Working at Welfare Evacuation Shelters during Natural Disasters and Multiple Disasters: A Qualitative Study"

_ijerph, 2022, doi:10.3390/ijerph192416610_

Round 1

Reviewer 1 Report

This is an intriguing study, and the authors used qualitative research methods to acquire unique data. In overall, the manuscript is nicely written and organized. However, I believe that the study contains certain shortcomings in various data analyses. Given these flaws, the work requires modest adjustments.

There were very few nurses who participated in this interview. How will it not cause any biasness in your study? There is a possibility that it could produce a biased outcome. What measures did you take to overcome this limitation?

Describe the relevance of this study and how it will be beneficial. The discussion part has very little explanation, which is inadequate. This section must be elaborated.

A total of 91 nurses 104 responded to our response sheet. Of these, 78 nurses declined to participate in the 105 interviews, and 13 nurses were willing to participate in the study. Of these, seven nurses 106

A question arises, Why did the majority of nurses refuse to participate in this interview? Knowing the cause for their unwillingness to participate in the interview would aid in resolving this problem.

Why authors have chosen to do interview-based qualitative research. Why were alternative types of qualitative research, such as case studies, not considered for this study?

Reviewer 2 Report

Thank you for reading this manuscript. The manuscript analyzes a sad disaster and the events experienced by the nurses who actively participated in its prevention. The topic would be extremely important, but we must recognize the severe limitations of the manuscript. It is impossible to draw far-reaching conclusions from the number of included participants; only two nurses were in each group. Unfortunately, it is impossible to conclude what happened 11 years ago from the extremely low responses. 

Reviewer 3 Report

The research topic is original, and the topic selection is judged to be appropriate. In addition, the summary and conclusion have been adequately described. However, it would be good to add descriptions of the parts contributing to practice, research, education, and policy reflection based on the results in the conclusion.

Since most of the references are more than five years old, it is necessary to review recent literature.
